# Identification of Bamboo Species Based on Extreme Gradient Boosting (XGBoost) Using Zhuhai-1 Orbita Hyperspectral Remote Sensing Imagery

**DOI:** 10.3390/s22145434

**Published:** 2022-07-20

**Authors:** Guoli Zhou, Zhongyun Ni, Yinbing Zhao, Junwei Luan

**Affiliations:** 1College of Tourism and Urban-Rural Planning, Chengdu University of Technology, Chengdu 610059, China; zhouguoli@stu.cdut.edu.cn (G.Z.); zhaoyinbing06@cdut.edu.cn (Y.Z.); 2College of Earth Sciences, Chengdu University of Technology, Chengdu 610059, China; 3Key Laboratory of National Forestry and Grassland Administration/Beijing for Bamboo & Rattan Science and Technology, Institute of Resources and Environment, International Centre for Bamboo and Rattan, Beijing 100102, China; junweiluan@icbr.ac.cn; 4School of Geography, Archaeology & Irish Studies, National University of Ireland, Galway (NUIG), H91 CF50 Galway, Ireland

**Keywords:** XGBoost, SAM, random forest, classification, Southern Sichuan Bamboo Sea

## Abstract

Mapping the distribution of bamboo species is vital for the sustainable management of bamboo and for assessing its ecological and socioeconomic value. However, the spectral similarity between bamboo species makes this work extremely challenging through remote sensing technology. Existing related studies rarely integrate multiple feature variables and consider how to quantify the main factors affecting classification. Therefore, feature variables, such as spectra, topography, texture, and vegetation indices, were used to construct the XGBoost model to identify bamboo species using the Zhuhai-1 Orbita hyperspectral (OHS) imagery in the Southern Sichuan Bamboo Sea and its surrounding areas in Sichuan Province, China. The random forest and Spearman’s rank correlation analysis were used to sort the main variables that affect classification accuracy and minimize the effects of multicollinearity among variables. The main findings were: (1) The XGBoost model achieved accurate and reliable classification results. The XGBoost model had a higher overall accuracy (80.6%), kappa coefficient (0.708), and mean F1-score (0.805) than the spectral angle mapper (SAM) method; (2) The optimal feature variables that were important and uncorrelated for classification accuracy included the blue band (B1, 464–468 nm), near-infrared band (B27, 861–871 nm), green band (B5, 534–539 nm), elevation, texture feature mean, green band (B4, 517–523 nm), and red edge band (B17, 711–720 nm); and (3) the XGBoost model based on the optimal feature variable selection showed good adaptability to land classification and had better classification performance. Moreover, the mean F1-score indicated that the model could well balance the user’s and producer’s accuracy. Additionally, our study demonstrated that OHS imagery has great potential for land cover classification and that combining multiple features to enhance classification is an approach worth exploring. Our study provides a methodological reference for the application of OHS images for plant species identification.

## 1. Introduction

Bamboo, which belongs to the families *Poaceae* and *Bambusoideae*, is widely distributed in tropical, subtropical, and temperate regions. Bamboo accounts for about 0.86% of the world’s total forest area and has increased by nearly 50% over the past three decades, mainly in China and India [1]. Bamboo forests in China are mainly distributed in subtropical regions, such as Fujian, Jiangxi, Zhejiang, Hunan, and Sichuan Provinces [2]. Interest has grown in the ecological and socioeconomic value of bamboo forests, as bamboo forests are efficient carbon sinks that play a critical role in mitigating climate change [3,4,5,6] and environmental restoration [7,8,9]. For instance, *Moso bamboo* forests have an especially high carbon sequestration potential [10,11]. Bamboo is also an important wood substitute and has contributed greatly to rural development, poverty reduction, and increased employment [12,13,14]. What is more, bamboo can be described as “the whole body is a treasure”. Bamboo shoots can be eaten fresh or processed into ready-to-eat bamboo shoots, magnolia slices, and so on; bamboo stalks are used for building and weaving bamboo utensils and handicrafts; branches can be used to make brooms; culm sheath is the raw material for weaving sacks, carpets, insoles, and papermaking [15]. *Phyllostachys edulis* is a dual-use bamboo species for shoots and timber. A *Phyllostachys edulis* forestland is most suitable for growing edible fungi, such as *Dictyophora*, which has high edible and medicinal value. *Bambusa emeiensis* and *Bambusa rigida* have high cellulose content, fiber lengths between broadleaf and coniferous forests, and strong toughness and good plasticity and are excellent raw materials of papermaking to replace wood [16]. Owing to the enormous usage value of these three types of bamboo, the local bamboo industry has been driven to flourish, which not only solves the employment problem of local residents, but also brings them considerable income. All these have greatly contributed to the implementation of the “rural revitalization strategy” formulated by the Chinese government. However, different bamboo species are currently difficult to extract, which needs to be explored. Thus, it is particularly important to accurately classify and map bamboo forests and bamboo species.

Traditionally, bamboo species identification relies upon field investigations, which are time-consuming, labor-intensive, expensive, and not suitable for rapid and frequent identification of bamboo forests at large spatial scales [17,18,19]. Spaceborne remote sensing technology is extensively used to identify land surface resources due to its ability to frequently observe the earth’s land surface at high spatial resolutions in near real time [20,21,22,23]. Multispectral and hyperspectral remote sensing technologies are often combined with LiDAR for vegetation mapping, with an accuracy rate of over 90.0% [24,25,26]. Although LiDAR can effectively assist in species identification, its use is limited by inaccessibility and high cost [27,28]. Synthetic aperture radar (SAR) is mainly used to identify forest types and is not used for species identification [19]. Individual trees and even leaves can be identified using high-resolution images [29], but the spectral curve signatures of individual trees are susceptible to differences in canopy illumination and background signal [30]. Hyperspectral remote sensing technology has quickly developed in recent years because of its ultrahigh spectral resolution, which can capture subtle differences in highly similar species. Furthermore, hyperspectral imagery is usually superior to multispectral imagery in species identification studies [31,32].

There is an abundance of hyperspectral satellites, such as the Project for On-Board Autonomy (PROBA), Sentinel-3A, Greenhouse Gas Satellite-Demonstrator (GHGSat-D), Aalto-1, GomSpace Express-4B (GomX-4B), Indian Mini Satellite-2 (IMS-2), and International Space Station (ISS), to name a few. There are also data products available from satellites that are out of service, such as Earth Observing-1 (EO-1) and Environmental Satellite (Envisat) [33]. China mainly operates three major satellite systems for land, ocean, and meteorology [34], and the hyperspectral satellites that are in orbit and widely used include GaoFen-5 (GF-5) [35,36,37,38], ZiYuan-1 02D (ZY-1 02D) [39], HuanJing-1A (HJ-1A) [40,41,42,43], and Orbita Hyperspectral (OHS) [44,45]. However, few hyperspectral satellites have been utilized for the identification studies of bamboo species. The OHS satellite has been broadly used due to its exceptional data quality since its launch in 2018. For instance, the spectral resolution is up to 2.5 nm, the spatial resolution is 10 m, the swath width is 150 km, the revisit period is 6 days for a single hyperspectral satellite, and the comprehensive revisit period is about 1 day for the eight hyperspectral satellites. The application of these data includes the identification of tree species [46], land cover [47], cotton [48], and wheat [49] with an accuracy of over 80.0%, which lays the foundation for the identification of bamboo species.

Different feature combinations play different roles in forestland identification, which are also an important basis for distinguishing classification methods of hyperspectral images. For instance, Du et al. [50] summarized that hyperspectral image classification schemes include three categories: classification directly using the original image, classification by band selection or feature extraction of the original image, and multidimensional feature classification by extracting spatial features from the original image or introducing auxiliary data. Zhang [51] grouped hyperspectral image classification methods into three categories: spectral feature classification, spatial and spectral feature classification, and multifeature fusion classification. However, it is challenging to identify bamboo forests and even bamboo species using remote sensing images due to the spectral similarity between bamboo species and between bamboo and other vegetation [52]. Therefore, we need to mine more feature information to distinguish the species.

Recent studies have shown that spectral features, vegetation indices, texture features, and topographic features, which are commonly used in image classification, can improve classification to varying degrees [53,54,55]. The spectral features are decisive features that discriminate between species. The vegetation indices can capture information on condition and are well correlated with vegetation coverage and biomass [56,57]. Topography is one of the most important factors that affect the distribution of vegetation species, which indirectly changes the distribution of vegetation via the redistribution of light and hydrothermal conditions [58,59,60,61]. The texture features contain information on land cover, and the extraction of this feature can assist in land cover identification. Computing texture features based on the gray level co-occurrence matrix (GLCM) is one of the most widely used methods [62,63]. However, bamboo is usually widely distributed in patches where the terrain is complex. Methods that only consider a single feature are not conducive to the accurate identification of bamboo species in mountainous areas.

The support vector machine (SVM) has often been used to identify bamboo species using hyperspectral data because it can effectively overcome the Hughes phenomenon in high-dimensional data [64]. For example, Chen et al. [65] used an ASD FieldSpec Pro FR spectroradiometer to obtain hyperspectral data for *Phyllostachys edulis*, *Phyllostachys violascens*, and *Bambusa multiplex* and used the SVM to identify the species with an average accuracy >90%. Chu et al. [66] carried out the identification of 12 species of bamboo leaves in Sichuan, Zhejiang, Yunnan, and Guangdong using near-infrared hyperspectral curves (900–1700 nm) and SVM, and obtained an average accuracy of >95%. Tao et al. [67] identified tree species in Gutianshan Nature Reserve, Zhejiang Province, using airborne hyperspectral data (AISA) and SVM, and the user’s accuracy (UA) for *Phyllostachys edulis* was 86.36%. Zhang et al. [68] extracted forest information from Huangfengqiao Forest Farm in Hunan Province based on the Hyperion imagery, using Mahalanobis distance, SVM, and SAM, and the user’s accuracy was 84.21%, 73.68%, and 63.16%, respectively. Liu et al. [69] identified a *Phyllostachys edulis* forest in Yong’an City, Fujian Province, by combining HJ-1A imagery with SAM and obtained a UA of 79.55%. Cai et al. [70] determined tree species in Longquan City, Zhejiang Province, using GF-2 images with XGBoost and SVM, and obtained UAs of 88.35% and 84.47% for *Phyllostachys edulis.* In the above studies, the identification of bamboo species using only the measured spectral curves can achieve an accuracy of more than 90%, which is not suitable for monitoring bamboo forests in large areas. Meanwhile, the classification accuracy based on SVM and hyperspectral images has large variability. This is because SVM often suffers from overfitting when dealing with high-dimensional and noisy datasets, while ignoring screening and evaluation features [71]. Furthermore, the binary nature of SVM will limit its application in the remote sensing field, which usually requires decomposing multiclass classification into binary classification [72].

However, SVM consumes substantial computer memory and computing time when dealing with high-dimensional data, and it is difficult to choose certain parameters [73]. Compared with SVM, XGBoost usually displays better performance in the training stage, which can run more steadily and has a faster computation speed [74]. The characteristics of this model could be outlined in two points: one is that it has a faster computation speed than other gradient boosting tools, and the other is that the model has an excellent performance in classification and regression modeling. Meanwhile, random forest, as an ensemble algorithm that is increasingly used in classification, can yield the importance of input variables according to the mean decrease in accuracy [75]. Random forest was applied in this paper to determine which feature variables have higher importance for classification [76]. SAM is commonly utilized for hyperspectral data classification, can handle high-dimensional data, and is insensitive to illumination. However, most of the above-mentioned studies only identified bamboo forests in broad categories or a single bamboo species and did not identify different bamboo species.

Here, we aimed to (1) explore the utility of OHS imagery in the identification of bamboo species, (2) demonstrate the methodological feasibility of the XGBoost models and SAM, and (3) determine uncorrelated feature variables that contributed significantly to classification. To achieve these aims, we first combined SAM and the measured spectra to extract the bamboo species from OHS imagery, and we verified the accuracy of our classifications with field observations. Second, we integrated four types of features and the XGBoost model to evaluate the effects of three feature variable combinations on the classification accuracy of bamboo species. Third, we determined which optimal feature variables were important and uncorrelated for classification accuracy using random forest and Spearman’s rank correlation analysis. Our classification results advance our knowledge of bamboo forest locations, coverage area, and type, which are vital for the sustainable management of bamboo and the accounting of vegetation carbon sinks.

## 2. Materials and Methods

### 2.1. Study Area

We selected the Southern Sichuan Bamboo Sea Scenic Area and its surroundings as the study area (a total area of about 780 km²), which is located at the junction of Changning County and Jiang’an County in Yibin City, Sichuan Province, with a coordinate range of 104°52′25″–105°16′3″ E, 28°23′24″–28°34′22″ N (Figure 1). The scenic area is a typical Danxia landform with an elevation of 400–1000 m in low mountains. The east side of the scenic area has low mountains, and the rest of the area are hills with an elevation of 200–500 m. The two counties where the study area is located are rich in bamboo resources according to the forestry survey data from the Sichuan Academy of Forestry. The bamboo species in the area are predominantly *Phyllostachys edulis*, *Bambusa emeiensis,* and *Bambusa rigida* (all with a total area greater than 40 km^2^), but *Pleioblastus amarus*, *Lingnania intermedia,* and *Dendrocalamus latiflorus* are also found. *Phyllostachys edulis* is mainly distributed in the Southern Sichuan Bamboo Sea Scenic Area. The elevation of the scenic area is significantly higher than that of the surrounding area, and its terrain fluctuates greatly. *Bambusa emeiensis* is primarily distributed in the mountainous area on the eastern side of the scenic area, and *Bambusa rigida* is mainly planted artificially and is mostly scattered in a small area that is relatively flat. The study area has a subtropical humid monsoon climate zone, where the annual average temperature is 14.5–18.0 °C, and the annual mean precipitation is 1200–2000 mm. The soil is mainly purplish soil and yellow soil, which is acidic and permeable. Benefiting from suitable water, heat, and soil conditions, the bamboo forest grows vigorously.

### 2.2. Materials

(1)Orbita hyperspectral (OHS) imagery

OHS is a group of commercial satellite constellations launched in China on 15 June 2017. Four of the hyperspectral sensors, OHS-2A, B, C, and D, were first launched on 26 April 2018 and have a swath width of 150 km and an orbital height of 520 km. The sensor generates images with a spectral resolution up to 2.5 nm and contains 32 bands in the wavelength range of 463–946 nm. We selected imagery from OHS-2C for our study, and the imaging time was 27 July 2020, Beijing time. The details of the imagery and the auxiliary data used are shown in Table 1.

The original OHS imagery is an L1B-level product, and users need to preprocess it to obtain a reflectance product, which involves three major steps. First, the original DN value was converted into radiance using the radiometric calibration coefficient, and then the actual surface reflectance was obtained using the FLAASH atmospheric correction model [77]. Second, orthorectification of the imagery was also required due to the topographic relief [78], which could reduce pixel displacement using Sentinel-2A and SRTM DEM as auxiliary data. Third, the SCS + C model [79] was used to perform topographic correction on the imagery, which could weaken the influence of topography on the radiation values of pixels by using AW3D30 DSM as auxiliary data. Finally, the desired study area was clipped from the imagery.

(2)Field data

Bamboo leaf collection was carried out from 21 July to 23 July 2021, which was aligned with the imaging season. The survey route was designed according to the forestry survey data and accessibility. Considering that the terrain in some areas was complex and difficult to reach, the nearest sampling point was selected to collect bamboo leaves. Figure 1c exhibits the final field survey route.

Synchronous measurements could not be completed because the average bamboo forest canopy height in the field exceeded 10 m. The bamboo leaves were removed, and the spectral information collection was conducted indoors. Our workflow was as follows: (1) the collected fresh bamboo leaf samples were bagged and sealed quickly; (2) the bags were labeled with the sample number, noting the bamboo species, coordinate location, and elevation with a record sheet as well; (3) we measured the spectral curves of bamboo leaves in a darkroom environment with SVC HR-1024i (high-resolution field portable spectroradiometer, its spectral range is from 350 to 2500 nm and has 1024 channels, produced by Spectra Vista Corporation, Poughkeepsie, NY, USA; for more information, please visit www.spectravista.com/ (accessed on 20 May 2021)); (4) we fixed the standard light source to match the instrument before the experiment and at a zenith angle of 30°; (5) the sensor probe was placed approximately 10 cm vertically above the sample surface with a 25° field of view; (6) five spectral curves were collected for each bag of samples, and the sensor was whiteboard-calibrated before collection; (7) the arithmetic average was calculated as the actual reflectance spectral curve after removing the abnormal curves in each bag; and (8) the measured spectral range was adjusted to match the spectral range of the imagery, and a spectral library was created.

(3)Accuracy assessment data acquisition

The accuracy of the classification results was evaluated using the field data. Considering the imaging time and local weather conditions, the field trip was conducted from 12 December to 14 December 2021. The bamboo morphology was relatively fixed during this period, which facilitated the identification of bamboo species in the field [15]. A total of 180 validation points were selected according to the final classification results and the actual terrain complexity of the study area, which included 62 for *Phyllostachys edulis*, 61 for *Bambusa emeiensis*, and 57 for *Bambusa rigida* (Figure 2). Photos of some field locations are demonstrated in Appendix A. These collection points would be used to generate confusion matrices and related evaluation indicators.

### 2.3. Methods

#### 2.3.1. Overview

Our overall workflow could be divided into three parts (Figure 3):(1)Data preprocessing is an essential basic work for subsequent identification of bamboo species, so the OHS imagery was subjected to radiometric calibration, atmospheric correction, orthorectification, and topographic correction, and then the forestry survey data were used to clip the desired study area.(2)Feature variables extraction and screening are key for processing high-dimensional data, which can effectively reduce information redundancy and improve computing efficiency. For SAM, the preprocessed imagery with 32 bands was used and combined with the measured spectral curves from SVC HR-1024i to perform spectral angle matching on the spectral curves of the reference samples. For XGBoost, spectral features, vegetation indices, and texture features were extracted from the preprocessed OHS imagery; topographic features were extracted from AW3D30 DSM. To quantify the main factors that affected classification and to minimize the effect of multicollinearity, an XGBoost model based on random forest [76] and Spearman’s rank correlation analysis was constructed.(3)Built on the results of variable screening, the SAM and the XGBoost model were used to classify bamboo species to obtain spatial distribution maps of bamboo species. The accuracy assessment and comparative analysis were finally implemented using field observations.

#### 2.3.2. Reference Samples Selection

The object-oriented method has been applied more often to the identification of specific land types than the pixel-based method, such as geohazards, impervious surfaces, and vegetation patches [80]. However, the premise of this kind of research is that the object can be effectively divided into independent polygons, and the object is usually expected to be countable. In addition, the object-oriented method often requires the image to have a high spatial resolution. Therefore, the pixel-based method is more suitable for bamboo forests with high-density continuous growth characteristics and images with medium spatial resolution.

Reference samples were selected with pixels as the minimum unit. Homogeneous pixels of *Phyllostachys edulis* (3098), *Bambusa emeiensis* (3086), and *Bambusa rigida* (3055) were selected on the preprocessed imagery. The reliability and accuracy of sample selection were ensured by combining the visual interpretation of remote sensing images with forestry survey data. Finally, the average spectral curves of the reference samples were calculated (Figure 4). It can be found that the signatures of spectral curves among bamboo species are very similar, especially for *Bambusa emeiensis* and *Bambusa rigida*.

#### 2.3.3. Spectral Angle Mapper

Spectral angle mapper (SAM) is a tool that can quickly identify the similarity between image spectrum and reference spectrum. The reference spectrum can come from the laboratory, the field measured spectrum, or the spectrum extracted from an image. This method requires that the image has been converted to “apparent reflectance”. SAM determines the spectral similarity by calculating the “angle” of two spectra regarded as vectors in N-dimensional space (N equal to the number of image bands). Equation (1) is the calculation formula of the “angle” [81]:(1)$$\alpha =\cos^{-1}\left(\frac{\sum_{i=1}^{nb}t_{i}r_{i}}{\sqrt{\sum_{i=1}^{nb}t_{i}r_{i}}\sqrt{\sum_{i=1}^{nb}t_{i}r_{i}}}\right)$$
where *α* is the “angle”, *nb* is the number of image bands, $t_{i}$ is the spectrum of class *i* in the image, and $r_{i}$ is the reference spectrum of class *i*.

The score ranking was acquired by calculating the “angle” between the reference sample spectrum and the reference spectrum of the spectral library. The higher the score of the reference spectrum, the higher the probability that the reference sample spectrum is this spectrum. The determination of the final spectrum also needs to be combined with prior knowledge. If the reference spectrum corresponding to the highest score is inconsistent with the prior knowledge, the reference spectrum with the second-highest score should be considered. Finally, the SAM classification was carried out with the selected reference sample spectra. The spectral “angle” threshold in the classifier was set to 0.1, and the pixels larger than this threshold would not be classified.

#### 2.3.4. Feature Variable Extraction and Screening

We selected a total of 48 feature variables, which included 32 spectral feature variables, 4 vegetation index feature variables, 4 topographic feature variables, and 8 texture feature variables.

(1)Spectral feature variables. The OHS imagery has rich spectral information. A total of 32 bands (B1 to B32) of the preprocessed OHS imagery were extracted as spectral feature variables.(2)Vegetation index feature variables. The vegetation index can be obtained by performing certain mathematical operations on multiple bands or band combinations of multispectral or hyperspectral remote sensing images. Four types of vegetation indices, namely, normalized difference vegetation index (NDVI), difference vegetation index (DVI), ratio vegetation index (RVI), and carotenoid index (CRI) [82,83,84,85], were chosen for feature analysis.(3)Topographic feature variables. The elevation of the study area (the data used are AW3D30 DSM) and its extracted slope, aspect, and slope position [86] were added to the feature variables.(4)Texture feature variables. The texture features of the GLCM [87] included mean, variance, homogeneity, contrast, dissimilarity, entropy, second moment, and correlation. Principal component analysis (PCA) was first performed on the preprocessed imagery, and then we selected the first principal component with a variance of 82.51% to calculate texture features. The processing window size of texture calculation was set to 3 × 3, and a total of 8 texture features were obtained.

Based on the feature variables extracted above, all variables were first normalized to 0–1, and then the variable value was extracted to each reference sample point. All reference sample points were used to establish the random forest model, which tends to obtain satisfactory results with default parameters [88], that is, *ntree* of 500 and *mtry* as the square root of the total number of feature variables. We determined the 15 most important variables. To minimize the influence of multicollinearity, Spearman’s rank correlation analysis [89] was performed on these 15 variables, and we excluded the variables with a correlation coefficient greater than 0.9 and lower importance [90], which allowed us to perform three classifications with different variable combinations: one with all variables (48 variables), the second with the top 15 most important variables (15 variables), and the third with only the most important and uncorrelated variables (7 variables).

#### 2.3.5. Extreme Gradient Boosting

Extreme gradient boosting, abbreviated as XGBoost, has been used frequently in various machine learning and data mining competitions since 2015. XGBoost is a scalable, portable, distributed gradient boosting library that provides gradient boosting decision trees that can solve many data science problems quickly and accurately, and is commonly used in Python and R language packages. The algorithm used by XGBoost is gradient boosting, which predicts the residual or error generated by the previous tree by creating a new tree and uses the gradient descent algorithm to minimize the loss caused by adding a new tree. The final prediction result is the sum of the predictions for all trees. The general formula for prediction at step *t* is as follows [91]:(2)$$f_{i}^{\left(t\right)}=\overset{t}{\underset{k=1}{\sum }}f_{k}\left(x_{i}\right)=f_{i}^{\left(t-1\right)}+f_{t}\left(x_{i}\right)$$
where $f_{t}\left(x_{i}\right)$ is the learner of step *t*, $f_{i}^{\left(t\right)}$ and $f_{i}^{\left(t-1\right)}$ are the prediction results of steps *t* and *t*−1, and $x_{i}$ is the input variable.

The model involves many parameters, most of which are about the bias-variance trade-off. The best model should take into account its complexity and predictive ability and prevent overfitting. There are usually two ways to prevent overfitting: the first is to directly control the complexity of the model through three parameters: *max_depth*, *min_child_weight*, or *gamma*; the second is to make the model training process robust to noise by increasing randomness. The adjustable parameters include the *subsample*, *colsample_bytree*, and *eta*. For more details and the calculation process of the XGBoost algorithm, see [91].

Three different variable combinations were fed into the XGBoost model after the extraction and screening of the feature variable. Then, XGBoost models with all the reference sample points were constructed using the *caret* package [92] in R statistical software and applied to the entire study area. A fivefold cross-validation was used to select the model containing the best combination of parameters [93]. The validation randomly divided the sample dataset into five subsets that were nonoverlapping and were approximately the same size. One subset was used as the validation set, and the remaining four subsets were used as the training set for modeling and calculating accuracy. This process was repeated five times, each time with a different subset as the validation set. Finally, the optimal model was chosen by averaging the accuracy of five results. The optimal parameters of the model under the three different variable combinations are illustrated in Table 2, and Appendix B displays the searching process of parameters.

#### 2.3.6. Accuracy Assessment

Accuracy assessment is an essential step in classification. The verification data used in the assessment came from the bamboo species validation points of the field trip. The confusion matrix and its common statistical indicators were selected to evaluate the classification performance of the SAM and XGBoost models [94,95]. Evaluation indicators included user’s accuracy (UA), producer’s accuracy (PA), overall accuracy (OA), kappa coefficient, and F1-score.

UA is the ratio of correctly classified pixels in a given class to all classified pixels in that class. PA is the ratio of correctly classified pixels in a given class to all reference pixels in that class. OA is the ratio of all correctly classified pixels to the sum of all pixels. Kappa analysis is a method used to quantitatively evaluate the agreement or precision between remote sensing classification maps and reference data, which is essentially measured by comparing the OA with the results of randomly assigning pixel categories. The F1-score is a weighted harmonic mean of UA and PA [96], which treats UA and PA as equally important.

## 3. Results

### 3.1. Spectral Angle Mapper

We calculated the “angle” and took the top five reference spectra by score (Table 3). Except for the spectrum of the *Bambusa rigida* sample, the spectra of both *Phyllostachys edulis* and *Bambusa emeiensis* samples agreed with the highest scoring reference spectra in the spectral library. The second-highest scoring reference spectrum was finally taken as the result of the spectrum of the *Bambusa rigida* sample according to prior knowledge. The results of the above spectral analysis proved that the selected reference samples were highly reliable and could be used for bamboo species identification.

Figure 5a is the confusion matrix based on SAM classification. According to UA, the order from low to high was *Phyllostachys edulis*, *Bambusa emeiensis*, and *Bambusa rigida*, and according to PA, the classification accuracy of *Phyllostachys edulis* was the highest (83.9%) and *Bambusa rigida* was the lowest (49.1%). The F1-scores of *Phyllostachys edulis* and *Bambusa emeiensis* were almost the same, while the F1-score of *Bambusa rigida* was lower. All bamboo species displayed a relatively well-balanced UA, while there were significant differences in PA. The error of omission was the highest for *Bambusa rigida*, while the error of commission was the highest for *Phyllostachys edulis*, which indicated that *Bambusa rigida* and *Phyllostachys edulis* were the most likely to be confused. It can be seen from the classification map (Figure 5b) that the classification results of *Phyllostachys edulis* were relatively continuous, but there were many *Phyllostachys edulis* patches in *Bambusa emeiensis*, and *Bambusa rigida* contained many *Bambusa emeiensis* and *Phyllostachys edulis* patches, and *Bambusa rigida* had a sporadic distribution.

### 3.2. Feature Variable Extraction and Screening

According to the growth and morphological characteristics of bamboo leaves, a total of 48 feature variables were selected for spectral features (1–32 bands), vegetation indices (NDVI, DVI, RVI, and CRI), texture features (mean, variance, homogeneity, contrast, dissimilarity, entropy, second moment, and correlation), and topographic features (elevation, slope, aspect, and slope position). More important variables need to be screened out, and unimportant variables need to be removed. By building a random forest model, the 15 variables with the highest contribution to the classification accuracy were selected (Figure 6). The previous step might have removed the multicollinearity between some variables, which did not completely guarantee the absence of multicollinearity. Although there were only 15 variables, some of them had a strong cross correlation. Therefore, Spearman’s rank correlation analysis was performed on these 15 variables to determine pairwise correlations, which revealed that the correlation coefficients within the green bands (B5, B6, B7) and within the near-infrared bands (B21, B22, B25, B26, B27, B28) were very high (Figure 7). DVI also had a very high correlation with the near-infrared bands.

Based on Spearman’s rank correlation coefficient (r < 0.9) and relative importance, seven important and uncorrelated variables were eventually retained, including B1, B27, B5, elevation, mean, B4, and B17, which are detailed in Appendix C. These variables were used to build one of the XGBoost models, with the purpose of reducing multicollinearity. After we compared the relative importance of these seven variables, the most important were the blue band B1 and the near-infrared band B27, which were almost equivalent. The importance of the green band B5 also exceeded 90. The importance of the elevation was close to that of the mean in texture features, while the importance of the green band B4 was substantially lower than that of the green band B5, and the red edge band B17 had the lowest importance.

### 3.3. Extreme Gradient Boosting

The best models were selected by the maximum accuracy derived from the fivefold cross-validation. It can be seen from Figure 8 that the accuracy of the optimal models under three different variable combinations decreased slightly with the reduction of the number of variables, while the time-consuming difference in the process of searching parameters was obvious. The time taken by the combination of important and uncorrelated variables was only half of that of all variables, which was 2 min less than that of the combination of important variables. The computer configuration used in the experiment included an Intel(R) Core (TM) i5-5257U CPU @ 2.70 GHz and 4 GB RAM.

Figure 9b and Figure 10b are the confusion matrix and classification map for XGBoost classification based on the combination of all variables. According to UA, the order from high to low was *Phyllostachys edulis*, *Bambusa rigida*, and *Bambusa emeiensis*. According to PA, the classification accuracy of *Bambusa emeiensis* was the highest (86.9%) and *Bambusa rigida* was the lowest (70.2%). The F1-score from high to low was *Phyllostachys edulis*, *Bambusa emeiensis*, and *Bambusa rigida*. There were significant differences in UA and PA across all bamboo species. The error of omission was highest for *Bambusa rigida*, while the error of commission was highest for *Bambusa emeiensis*, which indicated that *Bambusa rigida* and *Bambusa emeiensis* were the most likely to be confused.

An interesting phenomenon was found from the confusion matrices of XGBoost classifications of the other two variable combinations (Figure 9c,d): the rankings of UA, PA, and F1-score between bamboo species of these two classifications were consistent with the above classification, and the most easily confused bamboo species were also the same. However, the optimal evaluation indicators for a certain bamboo species varied among the different models. For example, the highest UA, PA, and F1-score for *Phyllostachys edulis* appeared in the XGBoost classification based on the combination of important and uncorrelated variables, the combination of important variables, and the combination of important variables, respectively. The classification maps (Figure 10c,d) show that the maps of the three XGBoost models were generally continuous and smooth, which agreed well with the forestry survey data.

### 3.4. Comparison of Classification Results

In terms of OA and kappa coefficient (Figure 11), the three XGBoost models were 8.8–10.0% and 0.13–0.16 higher than SAM, respectively, and the OA was 2.0–14.0% higher than similar studies [97,98,99]. The mean F1-score of the XGBoost models was 0.05–0.15 higher than that of SAM. From the classification results of single bamboo species, the UA of *Phyllostachys edulis* and *Bambusa rigida* in XGBoost models was higher than in SAM. The PA of *Bambusa emeiensis* and *Bambusa rigida* in XGBoost models was significantly higher than in SAM. The mean F1-score of the three bamboo species for the XGBoost models was higher when compared with SAM. Therefore, combining multiple feature variables helped to improve the classification accuracy, which can also be proved from previous studies. For example, Qi et al. [54] found that NDVI, NDMI, and texture features make bamboo forest classification more accurate and reliable. Li et al. [53] combined spectral bands, vegetation indices, texture, and topographic features to map the distribution of bamboo forests in Zhejiang from 1990 to 2014, and the OA was above 85%. Ghosh and Joshi [55] selected feature variables based on spectral bands, principal component variables, and texture features, and the PA could reach 82%.

Comparing the classification results of XGBoost under three different variable combinations (Figure 11), in terms of overall performance, the classification based on the combination of important and uncorrelated variables was slightly higher than that of the other two variable combinations in OA, kappa coefficient, and mean F1-score. Interestingly, the classification based on the combination of important variables performed slightly worse overall than the classification based on the combination of all variables. Judging from the classification results of single bamboo species, the classification based on the combination of important and uncorrelated variables manifested certain advantages in UA for *Phyllostachys edulis* and *Bambusa rigida*, and showed certain advantages in PA and F1-score for *Bambusa emeiensis*.

Figure 10 visualizes the results of SAM and three XGBoost classifications. It is obvious from the figures that the classification results of SAM were highly fragmented, while the classification results of the three XGBoosts were generally continuous and smooth, but the classifications based on the combination of all/important variables had local noise on the classification maps. In conclusion, the biggest difference between the two methods was that the overall classification performance of XGBoost and the identification performance of *Phyllostachys edulis* and *Bambusa rigida* were significantly better than SAM.

To further compare the regional distribution differences between the classification maps, they were superimposed to obtain Figure 12, where the purple areas represented the differences between two classification maps. The differences between SAM and the combination of the important and uncorrelated variables were the largest, and were mainly located in Taoping Township, Longtou Town, Hongqiao Town (the former was mostly *Bambusa emeiensis* and *Phyllostachys edulis*, while the latter was mostly *Bambusa rigida*), and Renhe Town (the former had a large number of *Phyllostachys edulis*, while the latter was mainly *Bambusa emeiensis*). The difference between the combination of all variables and the combination of important variables was concentrated in the transition zone, which was between *Bambusa rigida* and *Bambusa emeiensis* (Dipeng Town), and the classification results of the two were generally consistent. The differences between the combination of all/important variables and the combination of important and uncorrelated variables were similar, which were concentrated in the transition zone between *Phyllostachys edulis*, *Bambusa rigida* (Zhuhai Town, Longtou Town), *Bambusa rigida*, and *Bambusa emeiensis* (Dipeng Town).

## 4. Discussion

### 4.1. Spectral Angle Mapper

Generally, SAM achieved relatively ideal classification results, which indicated that *Phyllostachys edulis* was the best classified, *Bambusa rigida* was the worst classified, and *Phyllostachys edulis* and *Bambusa rigida* were the most likely to be confused. The most probable cause of confusion is that there were a large number of overlapping areas in the spectral signatures of *Phyllostachys edulis* and *Bambusa rigida* in the blue, green, red, and red edge bands. According to our field surveys, we found that *Bambusa rigida* was mostly scattered with a small area of distribution. Thus, the relatively poor classification results for *Bambusa rigida* were likely due to its sparsity and the resolution of the OHS imagery, which did not reach the submeter level. 

The analysis from the method itself lies in the fact that SAM does not consider sub-pixel values, so spectral mixing may become the biggest obstacle in the classification since most of the surface vegetation is heterogeneous [100]. Similarly, a related study found that the classification accuracy of SAM with Hyperion hyperspectral imagery, which has a spatial resolution of 30 m, often failed to achieve satisfactory results [101]. Therefore, spectral pixel unmixing technology could be considered in future research, which can weaken the adverse impact of mixed pixels on classification accuracy.

### 4.2. Feature Variable Extraction and Screening

Choosing the top 15 variables in terms of importance by building a random forest model might have removed some variables with multicollinearity, but the absence of multicollinearity was not fully guaranteed. Therefore, we conducted Spearman’s rank correlation analysis on these 15 variables, which indicated that there was a large amount of redundant information between the OHS imagery bands (especially between the near-infrared bands). In addition, DVI was highly correlated with the near-infrared bands, which was likely related to the use of the near-infrared band in the calculation of DVI. Thus, DVI was excluded from the optimal variable combination due to its high correlation with the near-infrared bands, which still had a high contribution to bamboo species identification. The best combination of feature variables was determined by feature variable extraction and screening: blue band B1, near-infrared band B27, green band B5, elevation, texture feature mean, green band B4, and red band B17. The important spectral bands we identified were consistent with the findings of other studies [88]. Thus, the role of spectral features in discriminating bamboo species, the vital influence of topographic features on vegetation distribution, and the supplementary potential of texture features for land class identification were decisive. 

Violin plots further demonstrate the validity of the screened variables (Figure 13). The plot depicts the location and relative amplitude of the different peaks when the data distribution is multimodal (more than one peak). The violin plots in Figure 13 present the distribution of each important and uncorrelated variable for the different bamboo species. All of these screened variables showed very large differences among bamboo species, which suggested that these variables had greater potential to distinguish the three species [102]. Elevation displayed a gradual change from high to low between *Phyllostachys edulis*, *Bambusa emeiensis*, and *Bambusa rigida*, and there were multiple peaks, which indirectly reflected the fluctuation in the terrain of the area, which was also consistent with our field results. Likewise, there were multiple peaks in the mean of the texture features, which indicated that there were variations in the canopy structure of bamboo species.

### 4.3. Extreme Gradient Boosting

Computational efficiency is often regarded as a key factor in evaluating classifier performance [103]. When hunting for the best combination of parameters by fivefold cross-validation, we found that the fewer the variables used, the shorter the runtime of the search process. Model accuracy declined only slightly with the reduction of the number of variables. This slight decline might be because there are few important feature variables in this dataset [91]. Thus, variable screening decreased computation time and model accuracy remained high.

In conclusion, the UA, PA, and F1-score of the XGBoost models under three different variable combinations exhibited the same ranking among bamboo species, which suggested that the classification results of the XGBoost models under different variable combinations were consistent and stable. Compared with similar studies, the UA of *Phyllostachys edulis* was 4.5–14.5% higher than that of SVM [67,69]. The highest UA of bamboo forest is about 7.0% and 15.5% higher than that of decision tree and random forest, respectively [54,99]. The OA of bamboo forest is comparable to that of the artificial neural network, but the UA is higher than that of the artificial neural network [104]. In this study, the classification performance of the three XGBoost models was substantially higher than SAM, and their classification results illustrated that *Phyllostachys edulis* was the best classified, *Bambusa rigida* was the worst classified, and *Bambusa rigida* was the most likely to be confused with *Bambusa emeiensis*. The most probable reason for the confusion was that the spectral signatures of *Bambusa rigida* and *Bambusa emeiensis* were very similar, and the spectral signature curves in the red edge bands were extremely close. Additionally, the similarity of the canopy structure of the two species might also cause confusion. We found that *Bambusa rigida* was mostly scattered and distributed in a small area, and so the resolution of the imagery may not resolve these small patches, which was likely responsible for its unsatisfactory classification.

### 4.4. Comparison of Classification Results

All three XGBoost models outperformed SAM in terms of the overall classification of bamboo species. This higher performance was most likely related to the fact that SAM only considered spectral features, while XGBoost incorporated more features. However, more features meant an increase in computational cost, so the computational burden was reduced by dimensional reduction. From the classification effect of a single bamboo species, SAM only manifested certain advantages in the UA of *Bambusa emeiensis* and the PA of *Phyllostachys edulis*, which might be because the reference spectra of the two bamboo species were consistent with the reference spectra in the spectral library (Table 3). Thus, XGBoost is likely a strong learner as it had better classification performance for hyperspectral images, which is in agreement with the results obtained by similar classification studies that used hyperspectral images [105]. From the results of classification visualization, the classification maps of XGBoost models are more in line with people‘s general cognition since Tobler‘s first law of geography points out that “everything is related to everything else, but near things are more related than distant things“ [106].

Overall, the XGBoost classification based on the combination of important and uncorrelated variables was slightly superior to the classification of the other two combinations of variables, and it also had certain advantages in terms of single bamboo species. Thus, data dimensionality reduction and the removal of multicollinearity by variable screening improved model classification performance, as has been demonstrated in related studies [90,107]. Moreover, it could be seen from the F1-score that the data processing method we proposed could better balance the UA and OA, which made the classification accuracy of single bamboo species higher. In the case of *Phyllostachys edulis*, the highest UA was 94.3%.

The accuracy of the classification of *Bambusa rigida* was the lowest for both classification methods. The most likely reason is that *Bambusa rigida* was scattered and had a small area of distribution, which made it difficult to obtain “pure” pixels when selecting its samples. The most effective way to address this problem in the future is to use hyperspectral images with higher spatial resolution or to explore the fusion of high spatial resolution images and hyperspectral images, such as pansharpening, Bayesian/MAP estimation, matrix factorization, and deep learning [108]. In addition, Lidar data, which can capture structural information, could also be considered to assist the class identification of hyperspectral data [109].

### 4.5. Limitations and Outlooks

The spatial resolution of OHS imagery limited the accuracy of bamboo species identification to some extent (especially *Bambusa rigida*), although the imagery was rich with spectral information. Furthermore, the highly fragmented distribution of *Bambusa rigida* itself hindered the accuracy of its identification. To further improve the classification accuracy, the fusion technology of hyperspectral and high spatial resolution images and the adoption of spectral pixel unmixing technology are the future research directions.

Our study area has complex terrain that includes mountains, hills, and plains; thus, the impact of different topographic correction models on the classification is also worth further exploring. In addition, cloudy and rainy weather and atmospheric and topographic effects make it difficult to obtain high-quality images of key stages of bamboo forest growth and are the limiting factors affecting research of mountain vegetation [110,111]. Only high-quality images for a single day were available for our study. However, these limiting factors could be addressed by shortening the satellite revisit period or by using drone photography.

The canopy reflectance among bamboo species will vary due to factors such as age, canopy closure, soil, water and heat, insect pests, and human disturbance [112,113]. Therefore, environmental variables that consider these factors can be added in future research, as these variables could help to improve the identification accuracy [114].

## 5. Conclusions

We explored the feasibility of using OHS imagery in bamboo species identification and the methodological feasibility of the XGBoost models and the SAM and determined the uncorrelated feature variables that contributed significantly to classification. Our accuracy assessments indicated that both the XGBoost models and SAM achieved satisfactory classification results, while the former had higher OA, kappa coefficient, and mean F1-score, which suggested that it was practical to identify bamboo species using OHS imagery. Additionally, the XGBoost model that used a combination of important and uncorrelated variables had the best classification performance, and the F1-score indicated that the data processing method we proposed could well balance the UA and PA.

Spectral similarity poses a challenge to bamboo species identification, but multifeature integration and screening are essential ways to improve accuracy. Feature variable screening that combines random forest with Spearman’s rank correlation analysis showed that the blue band B1, near-infrared band B27, green band B5, elevation, texture feature mean, green band B4, and red band B17 contributed more to the classification accuracy, and their correlation was lower. In general, the XGBoost model could capture information conducive to the identification of bamboo species, and species classification using OHS imagery is very promising. Our results have implications for regional bamboo industry planning, and our methods are valuable for species identification using hyperspectral images.

## Figures and Tables

**Figure 1 sensors-22-05434-f001:**
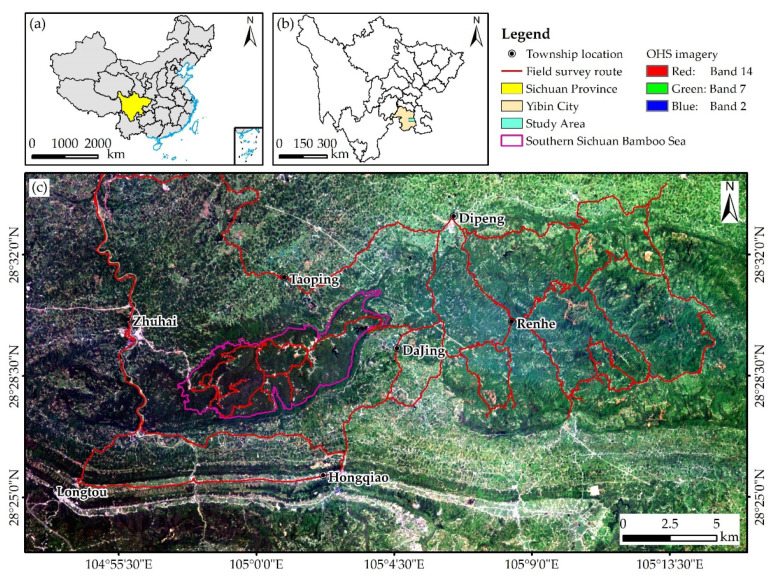
Location of the study area (**a**–**c**) and field survey route map (**c**).

**Figure 2 sensors-22-05434-f002:**
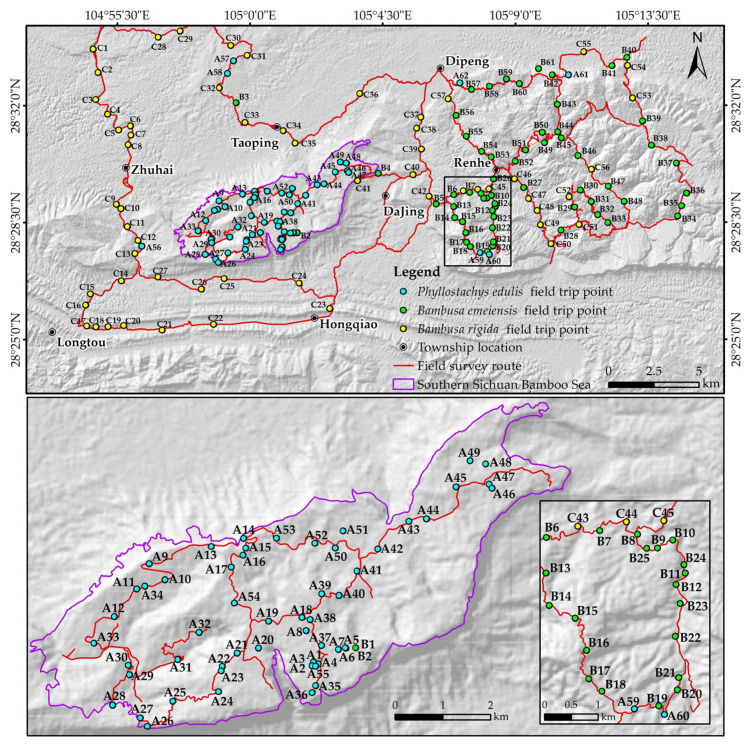
Map of field survey points and zoom-in maps.

**Figure 3 sensors-22-05434-f003:**
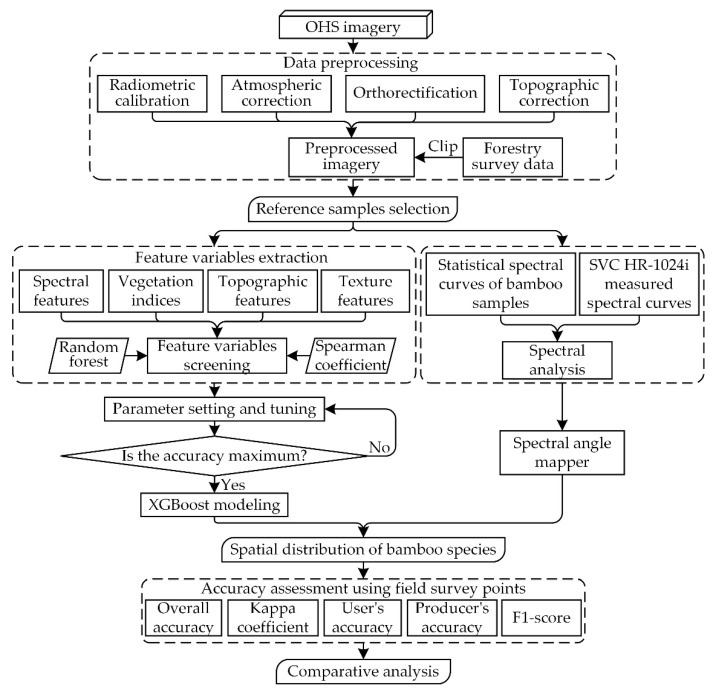
Bamboo species classification methodology and workflow.

**Figure 4 sensors-22-05434-f004:**
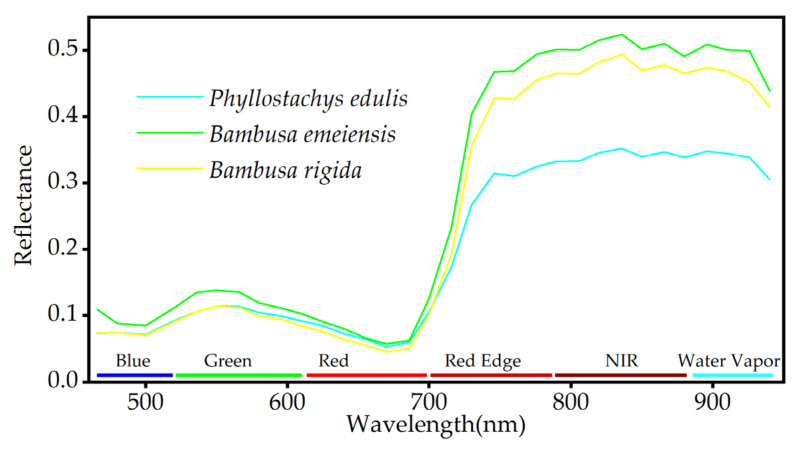
Average spectral signature curves of three bamboo species.

**Figure 5 sensors-22-05434-f005:**
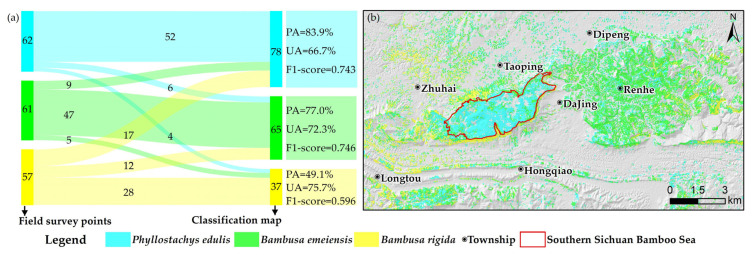
(**a**) Confusion matrix and evaluation indicators of SAM; (**b**) classification map of SAM.

**Figure 6 sensors-22-05434-f006:**
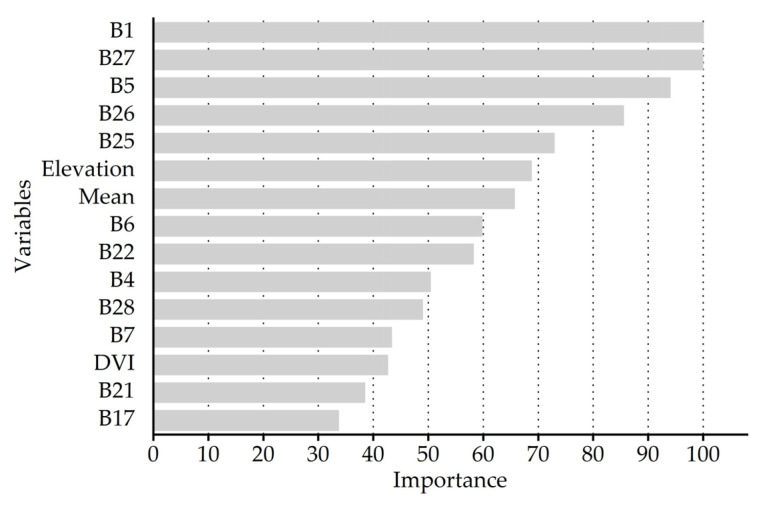
The 15 most important variables selected by the random forest model.

**Figure 7 sensors-22-05434-f007:**
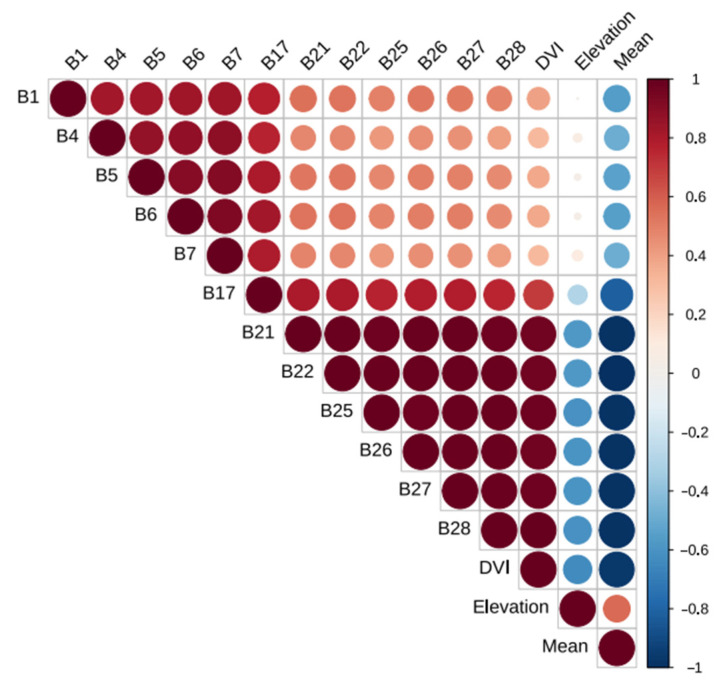
Spearman’s rank correlation coefficient for the 15 most important variables.

**Figure 8 sensors-22-05434-f008:**
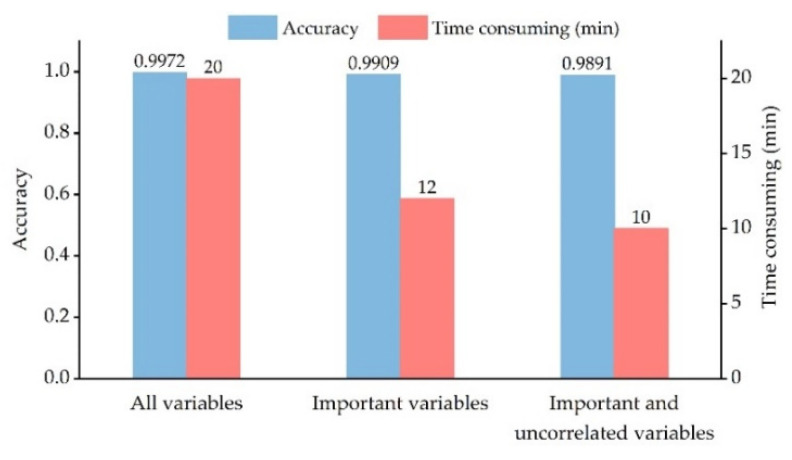
Parameter search results of XGBoost models under three variable combinations.

**Figure 9 sensors-22-05434-f009:**
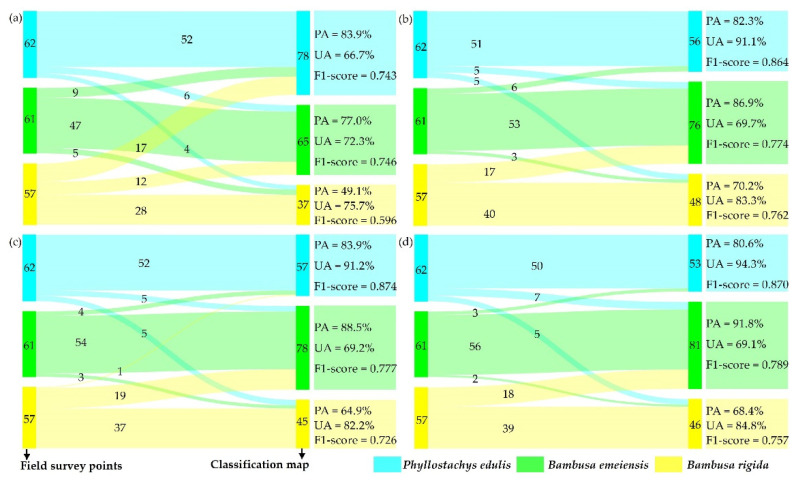
Confusion matrices and evaluation indicators: (**a**) SAM, (**b**) XGBoost (all variables), (**c**) XGBoost (important variables), and (**d**) XGBoost (important and uncorrelated variables).

**Figure 10 sensors-22-05434-f010:**
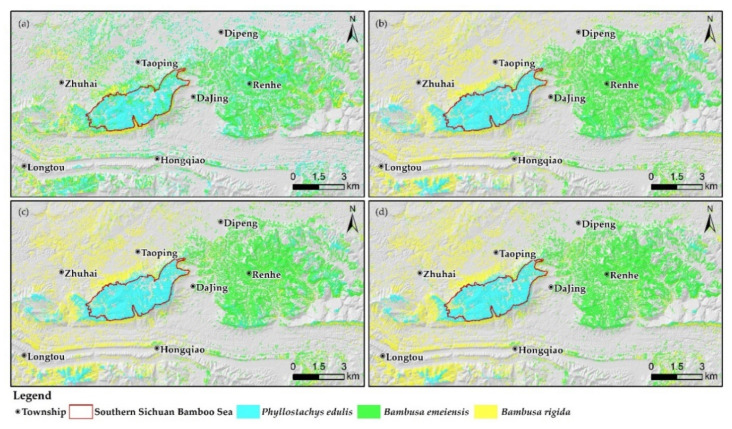
Classification maps: (**a**) SAM, (**b**) XGBoost (all variables), (**c**) XGBoost (important variables), and (**d**) XGBoost (important and uncorrelated variables).

**Figure 11 sensors-22-05434-f011:**
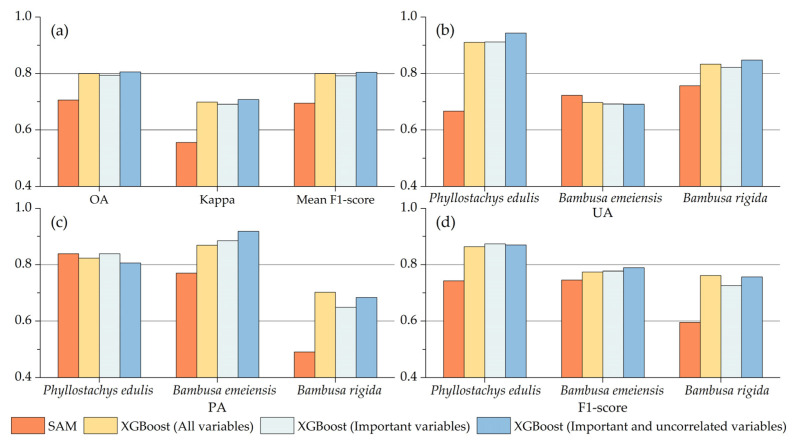
Comparison of evaluation indicators between SAM and three XGBoost classification results: (**a**) OA, kappa, and mean F1-score; (**b**) UA; (**c**) PA; and (**d**) F1-score of bamboo species.

**Figure 12 sensors-22-05434-f012:**
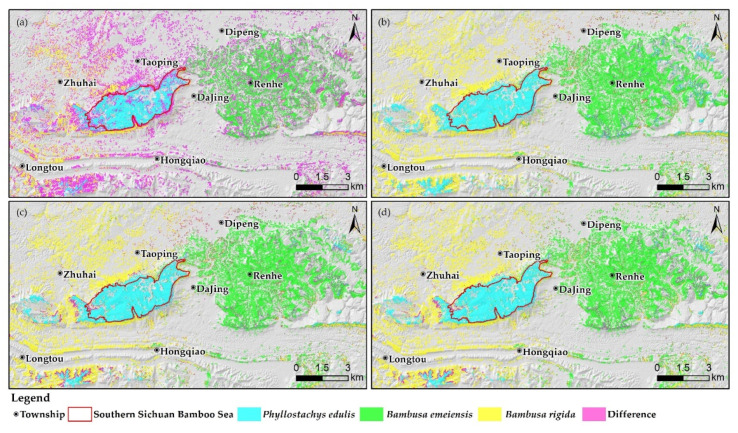
Comparison of differences in classification maps: (**a**) SAM vs. XGBoost (important and uncorrelated variables), (**b**) XGBoost (all variables) vs. XGBoost (important variables), (**c**) XGBoost (all variables) vs. XGBoost (important and uncorrelated variables), and (**d**) XGBoost (important variables) vs. XGBoost (important and uncorrelated variables).

**Figure 13 sensors-22-05434-f013:**
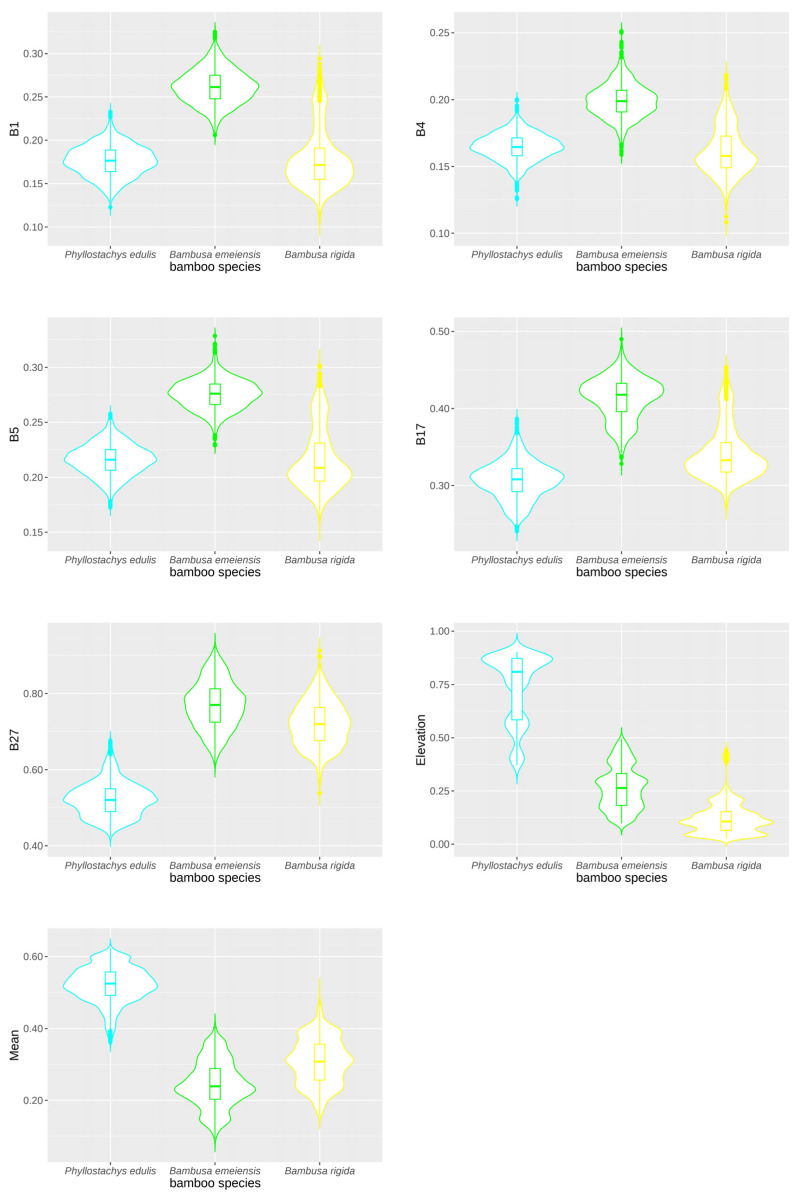
The distribution of values by bamboo species for the important and uncorrelated variables. Note: All feature variables are normalized to 0–1.

**Table 1 sensors-22-05434-t001:** OHS imagery and auxiliary data we used in this study.

Data Type	Acquisition Date	Spatial Resolution	Data Source
OHS imagery	27 July 2020	10 m	https://www.obtdata.com/ (accessed on 20 May 2021)
Sentinel-2A	15 August 2019	10 m	https://earthexplorer.usgs.gov/ (accessed on 20 May 2021)
SRTM DEM	11 February 2000	30 m	https://earthexplorer.usgs.gov/ (accessed on 21 May 2021)
AW3D30 DSM	May 2016	30 m	https://www.eorc.jaxa.jp/ (accessed on 21 May 2021)
Forestry survey data	15 October 2012	Vector data	Sichuan Academy of Forestry

**Table 2 sensors-22-05434-t002:** Optimal parameters of XGBoost models under three different variable combinations.

Variable Combination	*Nrounds*	*Max_Depth*	*Eta*	*Subsample*
All variables	400	5	0.3	1
Important variables	400	6	0.1	1
Important and uncorrelated variables	200	6	0.3	0.5

*nrounds*: the number of decision trees in the final model.

**Table 3 sensors-22-05434-t003:** The “angle” score of the reference spectrum and the reference sample spectrum.

*Phyllostachys edulis* Reference Spectrum	*Phyllostachys edulis* Sample Score	*Bambusa emeiensis* Reference Spectrum	*Bambusa emeiensis* Sample Score	*Bambusa rigida* Reference Spectrum	*Bambusa rigida* Sample Score
*Phyllostachys edulis01*	0.899	*Bambusa emeiensis04*	0.924	*Bambusa emeiensis04*	0.932
*Bambusa emeiensis04*	0.882	*Phyllostachys edulis01*	0.904	*Bambusa rigida05*	0.909
*Phyllostachys edulis02*	0.836	*Bambusa rigida05*	0.882	*Phyllostachys edulis04*	0.908
*Bambusa rigida01*	0.834	*Phyllostachys edulis04*	0.881	*Phyllostachys edulis02*	0.905
*Bambusa rigida05*	0.831	*Phyllostachys edulis02*	0.880	*Bambusa rigida04*	0.903

## Data Availability

Not applicable.

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
