# Peer review of "Identification of Bamboo Species Based on Extreme Gradient Boosting (XGBoost) Using Zhuhai-1 Orbita Hyperspectral Remote Sensing Imagery"

_sensors, 2022, doi:10.3390/s22145434_

Round 1

Reviewer 1 Report

This study achieved bamboo species classification based on hyperspectral remote sensing imagery using XGBoost. Spectra, topography, texture, and vegetation indices were included, and the results were compared with those obtained using SAM method which only takes spectra information into consideration. The results show improvement in accuracy, kappa coefficient, and mean F1-score.

Overall, it is a well written manuscript, and I suggest it to be accepted after the following issues are addressed.

1)     In the first paragraph of the introduction, while the importance of bamboo is well explained, the reason and importance of bamboo species classification is not enough. More details of the motivation of bamboo species classification are demanded.

2)     The first two sentences of paragraph two in section 2.1 are not relevant to a scientific manuscript.

3)     The instrument used (SVC HR-1024i) should be noted with its model, company, country, etc., according the requirements of the journal, even though it is a well-known instrument in the field.

4)     As claimed in the abstract, the manuscript tries to prove that by involving more feature variables, e.g., topography, texture, and vegetation indices, the classification can be improved. However, the results obtained from multiple feature variables and XGBoost were compared with the results obtained from spectra and SAM. In my point of view, the former is better compared with the results obtained from spectra only and XGBoost to make the statement more solid.

Reviewer 2 Report

This paper explored the feasibility of using OHS imagery in bamboo species identification, the methodological feasibility of the XGBoost models and the SAM, and determined the uncorrelated feature variables that contributed significantly to classification. The paper is well organized and the contributions are good. However, the following minor changes to be done before consider this paper for publication.

1. This paper does not provided sufficient literature study. Recommended to provide the literature along with the limitations of the existing works.

2. The Theoretical presentation of the work is poor. the authors can explain the proposed work theoretically.

Reviewer 3 Report

Brief summary

This study proposes a model to identify bamboo species using Hyperspectral Remote Sensing Imagery. Its main contributions consist in using feature variables such as spectra, topography, texture, and vegetation indices in a XGBoost model based on optimal feature variable selection.

Broad comments

The document is relatively easy to read and follow.

The English needs minor review.

The document is well supported with references.

The subject of the paper is interesting and with a great potential of application.

One of the weaknesses of this study is the lack of comparison of proposed method with several others state-of-the-art methods.

Specific comments

In the Abstract authors say that “…the XGBoost model achieved ideal classification results”. What do you mean by ideal results? Please clarify in the text.

At the end of first paragraph of Introduction section authors say that “…it is particularly important to accurately classify and map bamboo forests and bamboo species”. It is not clear from the text the importance of accurately classify and map bamboo forests. Please rewrite paragraph.

The caption on Figure 1 should be “…Location of the study area (a)(b)(c)…”

In subsection 3.2 authors should explain why were chosen 15 variables. Also authors should explain in more detail why seven variables were eventually retained.
